# Effects of Obstructive Sleep Apnea on Epilepsy, and Continuous Positive Airway Pressure as a Treatment Option

**DOI:** 10.3390/jcm11072063

**Published:** 2022-04-06

**Authors:** Hyunjin Jo, Sujung Choi, Dongyeop Kim, Eunyeon Joo

**Affiliations:** 1Department of Neurology, Neuroscience Center, Samsung Medical Center, Sungkyunkwan University School of Medicine, Seoul 06355, Korea; bell530@naver.com; 2Graduate School of Clinical Nursing Science, Sungkyunkwan University, Seoul 06355, Korea; sujungchoi@hanmail.net; 3Department of Neurology, Seoul Hospital, Ewha Womans University College of Medicine, Seoul 03760, Korea; hap2028@naver.com

**Keywords:** epilepsy, obstructive sleep apnea, polysomnography, CPAP

## Abstract

Purpose: To compare the characteristics of obstructive sleep apnea (OSA) between patients with epilepsy and patients without epilepsy and to investigate CPAP (Continuous Positive Airway Pressure) effect on seizures. Methods: Medical and polysomnography (PSG) data from 235 adult OSA patients with epilepsy (OE; 183 males; mean age, 49.8 years) and 268 age- and sex-matched OSA patients without epilepsy (OSE; 216 males; mean age, 51.3 years), obtained between March 2014 and May 2020 and housed in a database in a university-affiliated hospital, were retrospectively reviewed. All subjects completed surveys addressing comorbidities and medications, and sleep-related questionnaires including the Insomnia Severity Index, Epworth Sleepiness Scale, Pittsburgh Sleep Quality Index, and Beck Depression Inventory-II. Results: Compared with the OSE group, the OE group reported fewer sleep-related complaints and less severe OSA-related PSG parameters, with a lower apnea-hypopnea index (24.9 vs. 33.4 events/h; *p* < 0.003), arousal index (23.3 vs. 30.8 events/h; *p* < 0.001), and oxygen desaturation index (19.6 vs. 28.8; *p* < 0.002). The OE group had fewer smokers and lower alcohol consumption but a higher body mass index (27.0 vs. 25.9 kg/m^2^; *p* < 0.001). No correlations were observed between OSA-related PSG parameters and epilepsy-related factors, such as age at seizure onset, seizure type, frequency of seizures, presence of nocturnal seizures, and number of antiseizure medications, in the OE group. Patients with OE who demonstrated good compliance with CPAP therapy exhibited a decrease in seizure frequency. Conclusions: The OE group exhibited less severe disease characteristics than their age- and sex-matched OSE counterparts. Nevertheless, because the coexistence of OSA and epilepsy is high, CPAP therapy can reduce the frequency of seizures. Therefore, it is important to evaluate the presence of OSA in patients with epilepsy and to treat the conditions concurrently.

## 1. Introduction

Sleep disorders and epilepsy are both common conditions in the general population; as such, comorbidity and reciprocal influence are likely to exist. Epilepsy is the fourth most common chronic neurological disease, affecting approximately 70 million individuals worldwide [1], and sleep is a well-recognized, modifiable control measure for seizure control [2]. Various sleep disorders, such as obstructive sleep apnea (OSA), rapid eye movement (REM) sleep behavior disorder (RBD), and restless legs syndrome (RLS), can occur in patients with epilepsy [2,3]. OSA has been increasingly recognized as a significant comorbid condition in people with epilepsy (PWE) [4]. A recent meta-analysis reported that the prevalence of mild-to-severe OSA in PWE was 33.4% (95% confidence interval (CI) 20.8–46.1%), and PWE were more susceptible to OSA compared with healthy controls (odds ratio (OR) 2.36 (95% CI 1.33–4.18)) [5].

OSA is characterized by recurrent episodes/events of partial or complete pharyngeal collapse during sleep, causing sleep fragmentation and decreased oxyhemoglobin saturation [6], and is associated with extreme daytime hypersomnolence, cardiovascular morbidity, and mortality [7,8,9]. Among older adults with epilepsy, sleep apnea is associated with worsening seizure control or late-onset seizures [10]. Evidence from the past few decades suggests that an inactive lifestyle, obesity, and/or antiseizure medications (ASMs) contribute to the occurrence of OSA among PWEs [3,11].

However, few studies have explored sleep and respiratory characteristics on the basis of polysomnography (PSG) data and their relationship with seizures in PWE. Moreover, the effect of continuous positive airway pressure (CPAP) therapy on seizure outcomes in OSA patients with epilepsy (OE) has not been sufficiently explored.

We hypothesized that individuals with OE would have different PSG profiles or OSA-related symptoms compared with OSA patients without epilepsy (OSE), and that CPAP therapy, a treatment option for OSA, may improve seizure outcomes in those with OE. To evaluate this hypothesis, the present study aimed to characterize demographic characteristics, analyze and compare questionnaire results and PSG findings between the OE and OSE groups, and to estimate CPAP results in those with OE.

## 2. Methods

### 2.1. Subjects

Data from patients with epilepsy with newly diagnosed OSA were retrospectively included. Data were collected at the sleep center of a university hospital between March 2014 and May 2020. Individuals < 30 or >80 years of age; non-Korean-speaking foreigners; those with serious medical, mental, or neurological conditions, except for epilepsy; those with any sleep disorders, such as RBD, RLS, or narcolepsy confirmed by PSG; and shift workers were excluded.

During the recruitment period, 395 epilepsy patients underwent overnight PSG and were instructed to complete self-administered questionnaires before the sleep study. The study group comprised 296 patients diagnosed with OSA (apnea-hypopnea index [AHI] ≥ 5 events/h). Among them, 54 patients who were <30 years or >80 years of age, two unable to complete the questionnaires, three with comorbid psychiatric diseases, and two diagnosed with RBD were excluded (Figure 1). As the severity of OSA varies according to age and sex [12], 339 patients with newly diagnosed OSA without epilepsy (OSE), who were age- and sex-matched to an OE group during the same period, were enrolled. Among them, 71 were excluded due to the following factors: age < 30 years or >80 years (*n* = 42), neurodegenerative disease (*n* = 7), RBD (*n* = 16), narcolepsy (*n* = 5), and shift workers (*n* = 1). As such, a total of 235 patients with OE and 268 age- and sex-matched patients with OSA were included in this study.

### 2.2. Methods

This study was approved by the Institutional Review Board (SMC IRB No. 2021-02-023) and performed in accordance with the relevant guidelines and regulations of the Medical Ethics Committee while fulfilling ethical standards. Data obtained from medical records and self-administered questionnaires were retrospectively reviewed.

#### 2.2.1. PSG

A method previously published by the authors’ group for the recording and interpretation of PSG data was used [13].

#### 2.2.2. Clinical Information

Data were collected from surveys addressing basic demographic factors (age, sex, marital status, education level, occupation, alcohol consumption, and smoking) and medical comorbidities (hypertension, diabetes mellitus, cardiovascular disease, and dyslipidemia). Anthropometric data were obtained using standardized protocols for body mass index (BMI) and neck, waist, and hip circumferences.

Among those in the OE group, the following information was also collected from medical records: age at seizure onset; epilepsy classification; presence of nocturnal seizures; and number of ASMs in use (at the time of PSG). The categorization of ASMs according to the mechanism of action referred to previous studies [14]. Seizure frequency was defined as the average number of seizures per month in the baseline phase during the year before PSG.

Subjects who were prescribed CPAP were required to continue CPAP treatment for at least six months. Subject compliance and seizure frequency were assessed after six months of therapy. Data were reviewed for compliance with CPAP therapy. Good compliance with CPAP therapy was defined as use of the device for >4 h per night and for >70% of the observed nights based on machine downloads [15].

#### 2.2.3. Self-Reported Questionnaires

Daytime sleepiness: The Korean version of the Epworth Sleepiness Scale (ESS) was used to assess subjective daytime sleepiness [16]. An ESS score > 10 was considered to indicate clinically significant daytime sleepiness [16,17].

Insomnia severity: The Korean version of the Insomnia Severity Index (ISI) measures the subjective severity of insomnia symptoms and their consequences [18]. The ISI consists of seven items addressing current sleep problems and sleep-related distress during the previous two weeks. Scoring for each item ranged from minimal (0 points) to very severe (4 points), with a higher total score indicating more severe insomnia [19].

Sleep quality: The Korean version of Pittsburgh Sleep Quality Index (PSQI) measures the sleep quality and sleep disturbance during a 1-month period [20]. It consists of 19 items and is composed of 7 subscales (subjective quality of sleep, sleep latency, sleep duration, sleep efficiency, sleep disturbance, medication use for sleep, and daytime dysfunction). The total score ranges from 0–21, and higher scores indicate worse sleep quality [21].

Depression: Depressive symptoms were measured using the Korean version of the Beck Depression Inventory (BDI-II) [22]. It has a total of 21 items addressing depressive symptoms over the previous two weeks. Scoring for each item ranged from 0 (absence of symptoms) to 3 (most severe levels), and total scores ranged from 0 to 63, with a higher score reflecting a more depressed mood [23].

### 2.3. Statistical Analysis

All statistical analyses were performed using SPSS version 21.0 (IBM Corporation, Armonk, NY, USA) for Windows (Microsoft Corporation, Redmond, WA, USA). Differences noted with a two-tailed *p* < 0.05 were statistically significant. Descriptive data were expressed as mean ± standard deviation or frequencies and percentages. All continuous variables were analyzed using the independent *t*-test, and categorical variables were analyzed using the chi-squared test or Fisher’s exact test. Statistical analysis was performed with nonparametric measures using the Wilcoxon matched-pairs signed-rank test and Wilcoxon two-sample test. The OE and OSA groups were compared for categorical factors (generalized vs. focal epilepsy and presence vs. absence of nocturnal seizures). Spearman correlation coefficients were calculated to assess the correlation between demographic data, sleep questionnaires, PSG, and epilepsy-related factors in those with OE.

## 3. Results

### 3.1. Demographics and Subjective Sleep Questionnaires

In both groups, most of the subjects were middle-aged males (77.9% with OE, 80.6% with OSA). There was no statistical difference in the mean age of the OE and OSE groups (49.8 ± 11.9 years vs. 51.3 ± 11.0 years, respectively; *p* = 0.146).

The OE group had fewer years of education, a higher rate of unemployment, and a lower rate of both alcohol consumption and smoking compared with the OSE group. Comorbidities, however, were similar between the groups.

Insomnia symptoms, sleep quality, and daytime sleepiness were less severe in the OE group than in the OSE group (*p* < 0.001). Complaints of apnea/snoring and maintenance insomnia based on PSQI subitems were more frequent in those with OSE than those with OE. The primary sleep complaints in the OE group included snoring and increased napping. OE patients had higher depression scores (K-BDI-II) than OSE patients (14.5 ± 10.0 in OE vs. 12.3 ± 7.2 in OSA; *p* = 0.011) (Table 1).

In the OE group, the mean age at seizure onset was 31.0 ± 17.5 years and 83.0% of patients were diagnosed with focal epilepsy. Among focal epilepsy, lesional-epilepsy was more common than nonlesional epilepsy (65.1% vs. 34.9%), and temporal lobe epilepsy was more common than frontal lobe epilepsy (40.0% vs. 19.0%). The mean seizure frequency was 1.5 ± 8.3 episodes/month, and 47.7% of patients reported nocturnal seizures. The mean number of ASMs taken was 2.3 ± 1.7, and the most commonly used ASMs were voltage-gated ion channels. The details are summarized in Table 2.

### 3.2. Anthropometric and PSG Parameters

Both groups exhibited a higher percentage of N1 sleep and a lower percentage of N3 sleep; AHI and oxygen desaturation index (ODI) were higher in OSE than in OE. REM sleep percentage was relatively low in both groups but was statistically lower in those with OE. BMI was significantly higher in the OE group; however, AHI and ODI were lower in this group.

Arousal index (AI) was significantly higher in OSE due to a significantly higher proportion of respiratory arousals. The AI was lower in those with OE, and patients taking ≥ 2 ASMs exhibited a lower AI within the OE subgroup analysis. It was speculated that the lower AI observed in the OE group compared with the OSE group could be an effect of ASMs. Patients taking ≥ 2 ASMs had a higher percentage of N2 sleep than those with <2 ASMs. Details of this analysis are summarized in Table 3.

### 3.3. Analyses of Correlation between Epilepsy-Related Factors and Polysomnographic Parameters/Subject Questionnaires in OE 

In the OE group, epilepsy-related factors, such as age at seizure onset, epilepsy classification (regardless of lesional or nonlesional epilepsy, temporal lobe epilepsy, or frontal lobe epilepsy), frequency of seizure, and presence of nocturnal seizures, were not correlated with PSG parameters. The number of ASMs taken demonstrated a negative correlation with AI (rho = −0.220, *p* = 0.001). Taking ASMs with a mechanism of GABA inhibition showed a negative correlation with AHI (rho = −0.166, *p* = 0.011) and AI (rho = −0.221, *p* = 0.001).

Epilepsy-related factors were not correlated with ISI or PSQI scores. Age at seizure onset was negatively correlated with ESS (rho = −0.148, *p* = 0.030), and seizure frequency and the number of ASMs were positively correlated with the BDI score (rho = 0.184, *p* = 0.020 and rho = 0.228, *p* = 0.003, respectively). Taking ASMs having a mechanism of GABA inhibition showed a positive correlation with ESS (rho = 0.149, *p* = 0.027), ISI (rho = 0.157, *p* = 0.02), and BDI (rho = 0.219, *p* = 0.004), and taking ionotropic glutamate receptors showed a positive correlation with ESS (rho = 0.190, *p* = 0.005) (Table 4).

### 3.4. CPAP Adherence and Seizure Outcome

CPAP was prescribed to 134 of 235 OE patients, with a mean AHI of 26.7 ± 16.7 events/h. Of the 134 patients, 101 (75.45%) agreed to undergo CPAP therapy. Of the 134 patients prescribed CPAP, seizure outcomes for CPAP use were analyzed for the remaining 61, excluding the following: 56 who were seizure-free at baseline, seven with missing data regarding seizure frequency after CPAP use, five with <6 months of CPAP use, and five who were lost to follow up. Of the 61 patients, 50 underwent CPAP therapy and 11 did not (either CPAP intolerant or refused therapy). Of the 50 patients who used CPAP, 24 were adherent and 26 were nonadherent. In this study, seizure frequency before and after CPAP use in a group that used CPAP well (i.e., adherent to CPAP (*n* = 24)) and in a group that did not (no CPAP and nonadherent to CPAP (*n* = 37)) was analyzed. The seizure-free ratio and the 50% seizure reduction ratio did not differ significantly between the two groups; however, seizure frequency before and after CPAP use demonstrated a significant decrease only in the CPAP-adherent group. Although the sample size was small, the same results were obtained in an analysis of 16 patients, excluding 45 who had concomitant medication adjustment during the CPAP trial period (Figure 2).

In OSE group, the CPAP acceptance rate was 78.2%, of which 33.7% demonstrated good compliance with CPAP use.

## 4. Discussion

### 4.1. Prevalence of OSA in PWE

The prevalence of OSA ranged from 7.75% to 75.7%, with a high-level of heterogeneity reported among 19 previous studies. According to a recent meta-analysis, the prevalence of mild-to-severe OSA (AHI or RDI > 5) in PWE was determined to be 33.4% (95% CI 20.8–46.1%) [5].

In our study, we observed that 74.9% (296 of 395) of epilepsy patients who underwent full-night PSG testing were diagnosed with OSA (Figure 1). Approximately two-thirds (66.0%) of OE patients exhibited moderate-to-severe OSA, with a mean AHI of 32.9 ± 16.7 events/h. These relatively high percentages indicate that a proportion of the subjects studied had been referred for PSG due to clinical indications. Reasons for referral included excessive daytime sleepiness, suspected OSA, and characterization of nocturnal spells. In addition, their demographic profile (older, more males) may have contributed to a higher risk for OSA.

### 4.2. Characteristics of OE Compared to OSA

OSA affects epilepsy and epilepsy aggravates OSA. PWE may experience a higher frequency of OSA than the general population, and the severity of epilepsy can increase the risk for OSA [24]. Epilepsy affects OSA for several reasons and can exacerbate OSA. Frequent and extensive interictal epileptiform discharges and/or seizures alter upper airway control during sleep [11], and cessation of seizures and reduction of interictal discharges by successful epilepsy surgery may stabilize nocturnal sleep, which would result in a decrease in sleep-related respiratory events [25,26]. ASMs such as benzodiazepines, may worsen OSA or increase the risk for OSA by reducing upper airway muscle tone and ventilatory response to hypoxia [27,28,29,30]. Valproic acid may induce weight gain, which may worsen OSA symptoms [31].

However, contrary to our expectations, we observed that OE patients experienced fewer severe symptoms and signs related to sleep-disordered breathing than those with OSA. There are several possible explanations for these results. First, the OE group in this study had fewer complaints of OSA-related clinical presentations, such as snoring and apnea, than the OSE group. In the PSQI subitems, more patients in the OSE group complained of snoring and apnea than those with OE did. In particular, apnea is the most reliable indicator of OSA [32]. Patients with less severe OSA in the OE group may have been responsible for this finding. Second, alcohol consumption and smoking history are known risk factors for OSA [33,34,35]. In this study, the OE group exhibited a considerably lower ratio of alcohol consumption and smoking habits than the OSE group, which may be attributed to sleep hygiene education in the epilepsy clinic. Those with OE may avoid activities known to promote breakthrough seizures, many of which overlap with good sleep hygiene [36]. This finding suggests that lifestyle modifications, such as abstinence from alcohol and smoking, have a positive effect on OSA severity as well as seizure control. Third, the use of ASMs may stabilize sleep. As mentioned earlier, ASMs may exacerbate OSA but exert beneficial effects on seizure control. ASMs control seizures not only via direct effects on neuronal excitability but also through stabilization of sleep; reduction of sleep transitions; and, possibly, decreased sleep deprivation and increased sleep efficiency [37]. Our results showed that the group taking ≥ 2 ASMs exhibited a higher ratio of N2 sleep and a lower AI, especially respiratory AI, than those taking < 2 ASMs. OSA severity (such as AHI and ODI) did not differ according to the number of ASMs. In addition, in our results, taking ASMs with a mechanism of GABA inhibition, such as benzodiazepines, showed a negative correlation with AHI and AI. These drugs are traditionally not recommended in patients with OSA due to the concern of worsening OSA via pharyngeal muscle relaxation, increased apnea duration, and hypoxia [38], but a recent review suggested there was no deleterious effect of the hypnotics on the severity of OSA, as measured by AHI or ODI [39]. Collectively, ASMs may lead to sleep stabilization and have little effect on OSA severity.

### 4.3. CPAP as a Treatment Option for OE

The proposed mechanisms underlying the deleterious effects of OSA on epilepsy include sleep deprivation, sleep fragmentation, cerebral hypoxemia, decreased cardiac output, cardiac arrhythmias, autonomic instability, and increased sympathetic activity [40]. A retrospective review and prospective study reported that sufficient OSA treatment improves seizure control [41,42,43]. In a study from the United States that enrolled 41 epilepsy patients with OSA, seizures were better controlled in those who complied with CPAP use, whereas no significant difference in seizure frequency was observed in the noncompliant group [44]. According to a meta-analysis, patients treated with CPAP were shown to exhibit a higher seizure-free rate than those who were not treated with CPAP (42.6% vs. 15.4%; OR 4.03 (95% CI 1.15–14.1); *p* = 0.01). Patients treated with CPAP were also shown to have a high OR value, representing a 50% reduction in seizure rate (52.4% vs. 14.7%, OR 6.12 (95% CI 0.74–50.5); *p* = 0.09), although the difference was not statistically significant [5]. In our study, there was no difference in the seizure-free ratio or 50% seizure reduction ratio between the two groups; however, the change in seizure frequency before and after CPAP use demonstrated a significant decrease in the CPAP adherent group (Figure 2).

The present study had some limitations. First, OE patients were enrolled from an epilepsy clinic; thus, most did not initially appeal to the necessity of a sleep study. This resulted in less severe insomnia, less daytime sleepiness, and less poor sleep quality in OE than in OSE. Second, since this is a single-center study, the generalizability of our results is limited. In addition, 56 of 134 OE patients prescribed CPAP (41.8%) were seizure-free at baseline, which may have reduced the pool of patients available to assess the effect of CPAP on seizure outcomes. As ASMs were increased during the CPAP follow-up period in more than one-half of those with OE (45 of 61), the effects of CPAP or ASMs on seizure outcomes were not clearly delineated.

## 5. Conclusions

To the best of our knowledge, this is the first comparative study to compare sleep-related data between OE and OSE. Comorbidities, such as epilepsy and OSA, are not uncommon; however, the comorbidity of OSA and epilepsy has often been ignored. We found that patients with epilepsy may have a moderate-to-severe degree of sleep-related breathing disorders, although they rarely recognize sleep-related symptoms. In this case, it was confirmed that CPAP treatment reduced the frequency of seizures.

Thus, a comorbidity of OSA should be considered in epilepsy patients because treatment of OSA, such as CPAP, may be effective for seizure control.

## Figures and Tables

**Figure 1 jcm-11-02063-f001:**
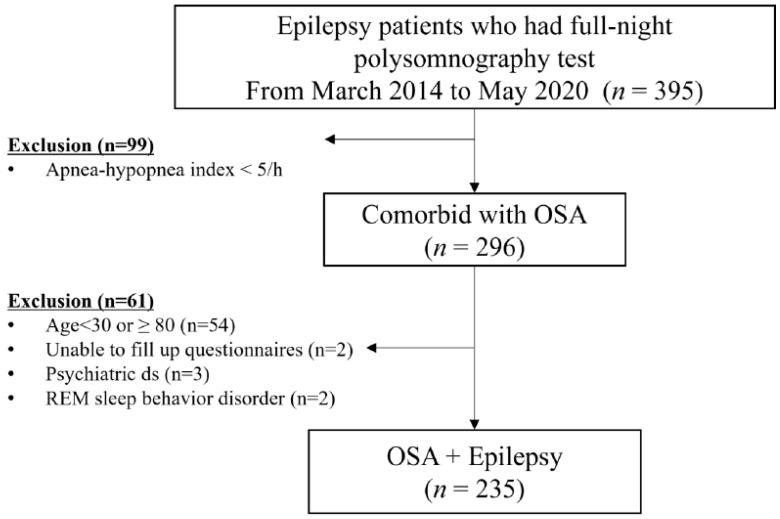
Enrollment log.

**Figure 2 jcm-11-02063-f002:**
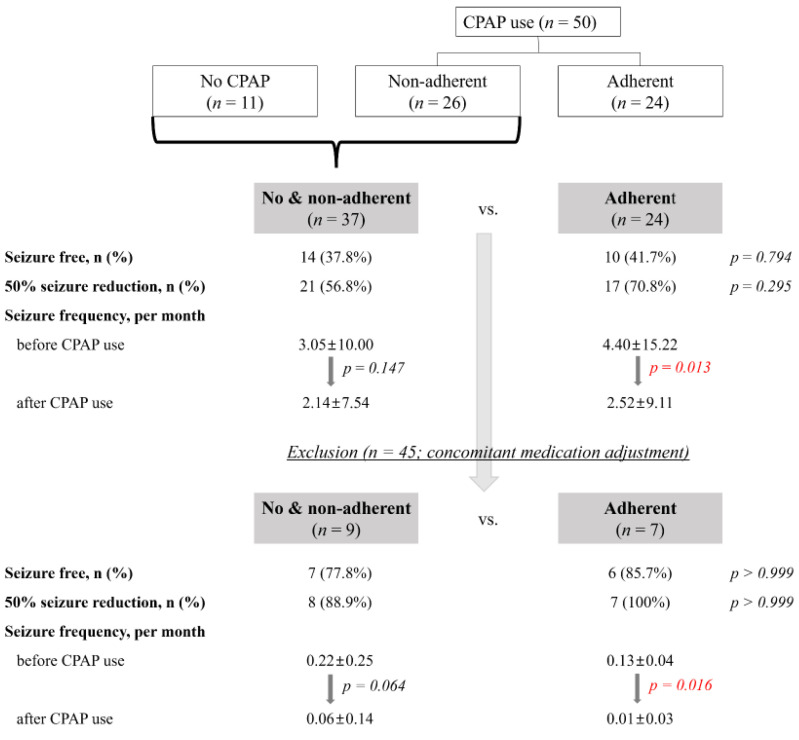
Flow diagram illustrating the use of continuous positive pressure (CPAP) therapy.

**Table 1 jcm-11-02063-t001:** Demographics and subjective sleep data.

Variables	Category	OSA Only (*n* = 268)	OSA with Epilepsy (*n* = 235)	*p*-Value
Age, years		51.3 ± 11.0	49.8 ± 11.9	0.146
Gender	Male	216 (80.6)	183 (77.9)	0.508 ^†^
Marital status	Single	28 (10.4)	48 (20.4)	0.002 ^†^
	Married	219 (81.7)	153 (65.1)	
	Divorced or widowed	19 (7.1)	17 (7.2)	
Education, *n* (%)	≤12 years	65 (24.3)	80 (34.0)	<0.002 *
	≥13 years	198 (73.9)	135 (57.4)	
Occupation ^†^	Yes	224 (83.6)	151 (64.3)	<0.001 *
	Students	3 (1.1)	1 (0.4)	
	No	32 (11.9)	64 (27.2)	
Alcohol, *n* (%)		175 (65.3)	89 (37.9)	<0.001 *
Smoking, *n* (%)	None	140 (52.2)	140 (59.6)	0.001 *
	Ex-smoker	81 (30.2)	35 (14.9)	
	Current	46 (17.2)	44 (18.7)	
Comorbidities	Hypertension	71 (26.5)	23 (9.8)	0.075
	Diabetes mellitus	24 (9.0)	11 (4.7)	0.541
	Cardiovascular disease	19 (7.1)	9 (3.8)	0.518
	Hyperlipidemia ^†^	11 (4.1)	4 (1.7)	0.482
Use of hypnotics, *n* (%)		23 (8.6)	4 (1.7)	0.056
Insomnia severity index		11.2 ± 6.3	8.8 ± 5.4	<0.001 *
PSQI		7.8 ± 3.4	6.0 ± 3.1	<0.001 *
5a. cannot get to sleep within 30 min		1.07 ± 1.15	0.93 ± 1.07	0.192
5b. wake up in the middle of the night or early morning		2.10 ± 1.03	1.59 ± 1.15	<0.001 *
5d. cannot breathe comfortably		1.18 ± 1.23	0.57 ± 0.97	<0.001 *
5e. cough or snore loudly		2.20 ± 1.10	1.55 ± 1.25	<0.001 *
5h. have bad dreams		0.69 ± 0.92	0.64 ± 0.87	0.576
ESS		10.7 ± 4.8	8.0 ± 4.4	<0.001 *
EDS (ESS > 10)		136 (50.7)	58 (24.7)	<0.001 *
K-BDI-II		12.3 ± 7.2	14.5 ± 10.0	0.011

PSQI, Pittsburgh sleep quality index; ESS, Epworth sleepiness scale; EDS, excessive daytime sleepiness; K-BDI-II, Beck depression inventory-II; ^†^ Analyzed by Fisher’s exact test. * *p*-value < 0.05.

**Table 2 jcm-11-02063-t002:** Epilepsy-related factors in patients with obstructive sleep apnea and epilepsy.

Variables	Category	Mean ± SD/*n* (%)
Age at seizure onset, year		31.0 ± 17.5
Epilepsy classification, *n* (%)	Generalized epilepsy	36 (15.3)
	Focal epilepsy	195 (83.0)
	lesional/nonlesional	127 (65.1)/68 (34.9)
	Temporal/Frontal	78 (40.0)/37 (19.0)
Frequency of seizure, /m		1.5 ± 8.3
Presence of nocturnal seizure, *n* (%)		112 (47.7)
Number of ASMs		2.3 ± 1.7
	0	29 (12.3)
	1	52 (22.1)
	≥2	150 (63.8)
Mechanisms of action of ASMs	Voltage-gated ion channels	172 (73.2)
	GABA inhibition	91 (38.7)
	Synaptic release machinery	94 (40.0)
	Ionotropic glutamate receptors	126 (53.6)

ASM, antiseizure medication; GABA, γ-Aminobutyric acid.

**Table 3 jcm-11-02063-t003:** Anthropometric and polysomnography parameters.

Variables	OSA Only (*n* = 268)	OSA with Epilepsy (*n* = 235)	*p*-Value	OSA with Epilepsy		
				ASM 0 or 1 (*n* = 81)	ASMs ≥ 2 (*n* = 150)	*p*-Value
BMI, kg/m^2^	25.9 ± 3.4	27.0 ± 3.6	<0.001 *			
≥25 kg/m^2^	162 (60.4)	158 (67.2)	0.069 ^†^			
Neck circumference-lying, cm	39.4 ± 3.4	39.2 ± 3.6	0.626			
Waist circumference, cm	92.7 ± 9.5	90.0 ± 19.1	0.065			
Hip circumference, cm	97.4 ± 6.3	97.8 ± 7.0	0.570			
Waist-hip ratio	0.95 ± 0.06	0.97 ± 0.57	0.008 *			
Total sleep time, min	371.1 ± 63.2	372.7 ± 59.0	0.773	377.0 ± 53.6	371.0 ± 61.7	0.458
Sleep latency, min	9.6 ± 14.3	10.1 ± 16.2	0.722	11.6 ± 22.1	8.9 ± 11.8	0.310
WASO, %	14.0 ± 9.4	14.3 ± 10.1	0.714	14.1 ± 8.8	14.4 ± 10.8	0.859
Sleep efficiency, %	84.3 ± 10.3	83.8 ± 10.8	0.501	84.0 ± 10.2	83.8 ± 11.2	0.916
Sleep stages						
N1 sleep, %	24.4 ± 14.1	20.5 ± 11.6	0.001 *	22.7 ± 12.7	19.3 ± 10.8	0.033
N2 sleep, %	52.2 ± 12.6	57.7 ± 11.7	<0.001 *	53.7 ± 12.3	60.0 ± 10.6	<0.001 *
N3 sleep, %	2.8 ± 5.2	4.5 ± 6.4	0.001	5.4 ± 5.8	4.0 ± 6.7	0.124
REM sleep, %	20.7 ± 6.9	17.3 ± 6.6	<0.001 *	18.3 ± 6.7	16.8 ± 6.5	0.097
AHI, /h	33.4 ± 21.3	24.9 ± 17.6	<0.001 *	28.5 ± 18.4	22.7 ± 16.5	0.015
OSA severity			<0.001 *			0.076
Mild (5 ≤ AHI < 15/h)	50 (18.7)	80 (34.0)		22 (27.2)	57 (38.0)	
Moderate (15 ≤ AHI < 30/h)	95 (35.4)	85 (36.2)		28 (34.6)	56 (37.3)	
Severe (AHI ≥ 30/h)	121 (45.1)	70 (29.8)		31 (38.3)	37 (24.7)	
Oxygen desaturation index	28.8 ± 21.4	19.6 ± 17.2	<0.001 *	22.3 ± 18.4	18.0 ± 16.0	0.083
Arousal Index, /h	30.8 ± 21.9	23.3 ± 12.1	<0.001 *	26.9 ± 13.2	21.4 ± 11.1	0.001
Respiratory AI, /h	22.8 ± 18.0	15.0 ± 12.7	<0.001 *	18.8 ± 15.0	12.9 ± 10.8	0.002
Spontaneous AI, /h	3.9 ± 4.0	4.1 ± 3.6	0.616	4.3 ± 3.7	4.0 ± 3.6	0.664

ASM, antiseizure medication; WASO, wake after sleep onset; REM, rapid eye movement; NREM, Non-rapid eye movement; AHI, apnea-hypopnea index; AHI_REM_, AHI during REM sleep; AHI_NREM_, AHI during NREM sleep; AHI_suprine_, AHI during supine position; AHI_lateral,_ AHI during lateral position; AI, arousal index. ^†^ Analyzed by Fisher’s exact test. * *p*-value < 0.05.

**Table 4 jcm-11-02063-t004:** Correlation between epilepsy-related factors and polysomnography parameters/subject questionnaires in patients with obstructive sleep apnea and epilepsy.

	AHI	ODI	AI	ESS	ISI	PSQI	BDI-II
Age at seizure onset	0.083	0.048	0.081	−0.148 *	−0.068	−0.026	0.014
Epilepsy classification							
Focal/Generalized	−0.106	−0.104	−0.041	0.023	0.072	0.023	0.028
lesional/nonlesional	0.055	0.064	0.057	−0.088	0.125	0.161	0.131
Temporal/Frontal	0.089	0.099	0.006	0.014	−0.097	0.073	0.109
Frequency of seizure (/month)	−0.054	−0.054	−0.014	−0.084	0.052	−0.034	0.184 *
Presence of nocturnal seizure	−0.004	−0.030	−0.015	−0.010	0.009	−0.066	−0.030
Number of ASMs	−0.109	−0.061	−0.220 *	0.129	0.144	0.036	0.228 *
Mechanism of action of ASMs							
Voltage-gated ion channels	−0.077	−0.040	−0.109	0.069	−0.013	−0.093	−0.063
GABA inhibition	−0.166 *	−0.109	−0.221 *	0.149 *	0.157 *	−0.038	0.219 *
Synaptic release machinery	−0.077	−0.074	−0.110	−0.023	0.069	−0.035	0.121
Ionotropic glutamate receptors	0.010	0.015	−0.100	0.190 *	0.092	0.013	0.137

Spearman rho values are presented; * *p*-value < 0.05; ASMs, antiseizure medications; GABA, γ-Aminobutyric acid.

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
