# Peer review of "Effects of Obstructive Sleep Apnea on Epilepsy, and Continuous Positive Airway Pressure as a Treatment Option"

_jcm, 2022, doi:10.3390/jcm11072063_

Round 1

Reviewer 1 Report

1. In general: The abbreviation "OSA" for the group of patients with OSA but without epilepsy does not seem ideal to me - among other things, because those with epilepsy also have OSA.  

2. Section 1: "Sleep disorders and epilepsy are both common conditions in the general population; as such, comorbidity and reciprocal influence are likely to exist." - I don't see why reciprocal influence should follow from co-existence. 

3. Section 1: "[...] sleep hygiene is a well-recognized, modifiable control measure for seizure control." - Sleep hygiene" is usually understood to mean "good sleep habits" - and there is no mention of this in the cited article.

4. Section 2.2.2: "Presence of nocturnal seizures": During the year before PSG?

5. Section 2.2.2: Has the type of antiseizure medication been documented? (Influence on sleep, body weight,...)

6. Section 3.2: "[...] OSA was more severe in those with OE" - is that correct?

7. "Arousal was significantly higher in patients with OSA [...]" - what do you mean? 

8.  Section 3.4: "[...] significant decrease only in the CPAP adherent group." - Can you give the numbers - and wouldn't figure 2 fit better here? Do you know how much nocturnal seizures decreased? Do you know whether quality of life changed?

9. Section 4.2: "OSA affects epilepsy and epilepsy aggravates OSA" - very superficial statement. 

10. "Lifestyle modifications [...] have a positive effect on OSA severity as well as seizure control."-  You did not examine the latter part, right? 

11. You suggest that ASM can stabilize sleep - does reference [4] really imply this?

Author Response

Comment 1:

In general: The abbreviation "OSA" for the group of patients with OSA but without epilepsy does not seem ideal to me - among other things, because those with epilepsy also have OSA.  

Response to comment 1:

We thank the reviewer’s insightful comment. We accept the reviewer's opinion and changed the abbreviation as follows.

OSA with epilepsy à OE vs. OSA without epilepsy à OSE

Comment 2:

In Section 1: "Sleep disorders and epilepsy are both common conditions in the general population; as such, comorbidity and reciprocal influence are likely to exist." - I don't see why reciprocal influence should follow from co-existence. 

Response to comment 2:

Since sleep disorders and epilepsy are both common disorders, it is highly likely that the two disorders are present in the same person at the same time, which means that an interaction between the two disorders is possible.

Comment 3:

In Section 1: "[...] sleep hygiene is a well-recognized, modifiable control measure for seizure control." - Sleep hygiene" is usually understood to mean "good sleep habits" - and there is no mention of this in the cited article.

Response to comment 3:

Thank you for your helpful comment. This is our mistake. We are not trying to explain "sleep hygiene," but rather the importance of "sleep". We change it to “sleep”, not “sleep hygiene”.

Comment 4:

Section 2.2.2: "Presence of nocturnal seizures": During the year before PSG?

Response to comment 4:

The presence of nocturnal seizures was not investigated for a set period. Nocturnal seizures were considered to be present if nocturnal seizures were reported during patient interviews and medical chart reviews.

Comment 5:

In Section 2.2.2: Has the type of antiseizure medication been documented? (Influence on sleep, body weight,...)

Response to comment 5:

ASMs were classified into four types according to the mechanism of action: voltage-gated ion channels, GABA inhibition, synaptic release machinery and ionotropic glutamate receptors (Reference). In the correlation analysis between this classification and sleep related parameters, ASMs with a mechanism of GABA inhibition showed negative correlation with AHI and AI and positive correlation with ESS, ISI, and BDI. Ionotropic glutamate receptors showed a positive correlation with ESS. However, ASMs acting by modulating voltage-gated ion channels and ASMs acting through interaction with elements of the synaptic release machinery did not show any correlation with sleep-related parameters. These contents were added to the text and tables (2 & 4).

Reference: Rogawski MA, Löscher W, Rho JM. Mechanisms of Action of Antiseizure Drugs and the Ketogenic Diet. Cold Spring Harb Perspect Med. 2016;6(5):a022780. Published 2016 May 2. doi:10.1101/cshperspect.a022780

(In the revised Page 5, line 16) The categorization of ASMs according to the mechanism of action referred to previous studies.

(In the revised Page 10, line 16) The most commonly used ASMs are voltage-gated ion channels.

(In the revised Page 14, line 4) Taking ASMs with a mechanism of GABA inhibition showed a negative correlation with AHI (rho = -0.166, p = 0.011) and AI (rho = -0.221, p = 0.001).

(In the revised Page 14, line 9) Taking ASMs having a mechanism of GABA inhibition showed a positive correlation with ESS (rho = 0.149, p = 0.027), ISI (rho = 0.157, p = 0.02), and BDI (rho = 0.219, p = 0.004), and taking ionotropic glutamate receptors showed a positive correlation with ESS (rho = 0.190, p = 0.005).

(In the revised Page 18, line 18) In addition, in our results, taking ASMs with a mechanism of GABA inhibition, such as benzodiazepines, showed a negative correlation with AHI and AI. These drugs are traditionally not recommended in patients with OSA due to the concern of worsening OSA via pharyngeal muscle relaxation, increased apnea duration, and hypoxia [28], but a recent review suggested there was no deleterious effect of the hypnotics on the severity of OSA, as measured by AHI or ODI [29].

Comment 6:

  1. Section 3.2: "[...] OSA was more severe in those with OE" - is that correct?

Response to comment 6:

Indices related to the severity of OSA such as AHI and ODI were higher in the OSE than in OE. Since that sentence may be confusing, I will amend the sentence as follows:

(In the revised Page 12, line 7) AHI and ODI were higher in OSE group than in OE group.

Comment 7:

  1. "Arousal was significantly higher in patients with OSA [...]" - what do you mean? 

Response to comment 7:

This sentence explains that “OSA patients without epilepsy” have a higher arousal index than “OSA patients with epilepsy”. As the reviewer pointed out in Point 1, it seems that the use of the abbreviation is incorrect, causing confusion. I changed the abbreviation as defined in Response1 and modify the sentence as follows.

(In the revised Page 12, line 10) Arousal index (AI) was significantly higher in OSE.

Comment 8:

In Section 3.4: "[...] significant decrease only in the CPAP adherent group." - Can you give the numbers - and wouldn't figure 2 fit better here? Do you know how much nocturnal seizures decreased? Do you know whether quality of life changed?

Response to comment 8:

In the no CPAP & non-adherent to CPAP use group, it decreased from 3.05±10.00 to 2.14±7.54 (p=0.147), whereas in the adherent to CPAP use group, it decreased from 4.40±15.22 to 2.52±9.11 (p=0.013). As the reviewer pointed out, I'll reposition Figure 2 after this section.

Since it is a retrospective study based on chart review, it was not possible to confirm whether there was a decrease in nocturnal seizures before and after CPAP use or the change in quality of life. We agree that further studies are needed to determine whether the use of CPAP reduces nocturnal seizures.

Comment 9:

In Section 4.2: "OSA affects epilepsy and epilepsy aggravates OSA" - very superficial statement. 

Response to comment 9:

This expression was used to emphasize the interaction between OSA and epilepsy. The statement itself is a superficial statement, but it is explained in detail later. Section 4.2 describes how epilepsy affects OSA, and section 4.3 describes how OSA affects epilepsy.

Comment 10:

"Lifestyle modifications [...] have a positive effect on OSA severity as well as seizure control."-  You did not examine the latter part, right? 

Response to comment 10:

This means that the seizures that can be provoked by alcohol and smoking can be controlled through lifestyle modifications.

Comment 11:

You suggest that ASM can stabilize sleep - does reference [4] really imply this?

Response to comment 11:

As the reviewer pointed out, the reference is misquoted. The study I intended to cite reported that improved sleep efficiency and decreased arousals from sleep on seizure-free nights. I will edit the Reference (I'm quoting [27], not Reference [4], here).

Reference [27] Touchon J, Baldy-Moulinier M, Billiard M, Besset A, Valmier J, Cadilhac J. Organisation du sommeil dans l'épilepsie récente du lobe temporal avant et après traitement par carbamazépine [Organization of sleep in recent temporal lobe epilepsy before and after treatment with carbamazepine]. Rev Neurol (Paris). 1987;143(5):462-7. French. PMID: 3659725.

Reviewer 2 Report

First of all, I would like to thank the authors for the chosen topic, which is relevant to the clinical practice of multiple medical specialties (primary care, neurology, geriatrics, pulmonology...). I will now go on to list aspects that I consider could potentially be improved and criticisms that I hope will be constructive: 

1) Epilepsy is NOT a single pathology. The correct thing would have been to choose a type of epilepsy, at least idiopathic or genetic generalized epilepsy, which in turn are more vulnerable to sleep disruption. In case of choosing focal epilepsies, it would be necessary to choose between lesional or non-lesional and preferably also to choose by location of the ictal focus. Frontal focal epilepsies have a higher prevalence of ictal and interictal epileptiform activity during sleep than temporal focal epilepsies. 

2) It is not clear the reason for referral for PSG in both subjects with and without epilepsy. It appears that there may be a subject selection bias in both groups that makes it difficult to extrapolate the results. 

3) The high percentage of subjects with epilepsy with 2 or more ASMs is surprising, more than expected a priori, and the same for seizure control (subotpimal to say the least). It is also impressive that it does not correctly reflect the population with epilepsy. 

4) It would be interesting NOT only to evaluate according to the number of ASMs but also according to their type (ideally by mechanism of action but at least according to generation). Some of them should have a less neutral impact on the structure of sleep, others could aggravate cardiovascular risk factors which in turn are risk factors for the development of OSA,....

5) No reference is made to the detection of intercritical epileptiform activity and/or electroclinical seizures in the group with epilepsy. It would be interesting to know whether apneic awakenings are associated with epileptiform abnormalities or not. This would further support the potential use of CPAP as a further added therapy in the course of subjects with epilepsies and OSA.

6) Ideally, you and myself would have liked to have longitudinal PSG and not just longitudinal sleep questionnaires to assess impact on sleep quality with respect to CPAP use in population with and without epilepsy. In case of technical or economic limitations for its realization, at least I believe that an actigraphy should be postulated. 

7) The subjects with epilepsy + OSA who complete treatment and follow-up with CPAP are very few and the statement that it reduces the frequency of seizures but do not increase the percentage of seizure-free subjects seems to be related to this sample size and a limited follow-up in time (I think that a one-year analysis could have been of more interest). 

8) To evaluate to provide information on intecritical epileptiform activity and the presence of electroclinical seizures and its relationship with AHI in the PSG. 

9) It would be of interest to evaluate the impact not only on seizure control and self-administered scales of sleep quality and depressive symptoms, but also on other neuropsychiatric symptoms (for example, using the NPI questionnaire) and the impact on global cognitive level (at least with MoCA or MMSE type test) to be performed pre and post-CPAP in both groups.

10) Consider changing the location of Figure 2, which I do not think it would be the most appropriate to place it in the discussion section.

I am aware that this is a retrospective study and that this entails limitations that cannot be rectified, but given the clinical importance of the question posed and the practical applicability of the results of the study, I believe it is essential to demand maximum rigor. 

Author Response

General comment:

First of all, I would like to thank the authors for the chosen topic, which is relevant to the clinical practice of multiple medical specialties (primary care, neurology, geriatrics, pulmonology...). I will now go on to list aspects that I consider could potentially be improved and criticisms that I hope will be constructive: 

We sincerely appreciate the constructive comments of reviewer that were helpful in improving the quality of our manuscript, and we have revised our manuscript accordingly. All the changes have been incorporated in the manuscript and have been highlighted (highlighted version of the revised manuscript). Kindly find our point-by-point responses (in blue) to the reviewer’s comments below.

Reviewer 1 pointed out that it was inappropriate to use the abbreviation "OSA" for a group of patients with OSA but without epilepsy, and this was changed as follows.

OSA with epilepsy à [OE] vs. OSA without epilepsy à [OSE]

Comment 1:

Epilepsy is NOT a single pathology. The correct thing would have been to choose a type of epilepsy, at least idiopathic or genetic generalized epilepsy, which in turn are more vulnerable to sleep disruption. In case of choosing focal epilepsies, it would be necessary to choose between lesional or non-lesional and preferably also to choose by location of the ictal focus. Frontal focal epilepsies have a higher prevalence of ictal and interictal epileptiform activity during sleep than temporal focal epilepsies. 

Response to comment 1:

I agreed with the reviewer's point and re-analyzed the information that can be found through the chart review. Unfortunately, it was not possible to determine whether generalized epilepsy was idiopathic or genetic. Focal epilepsy (n=195) was classified as lesional (n=127, 65.1%) or non-lesional (n=68, 34.9%), and whether ictal focus was temporal (n=78, 40.0%), or frontal (n=37, 19.0%) could be investigated. However, these classifications did not show correlation with PSG parameters such as AHI, ODI, and AI. These contents were added to the text and tables (2 & 4).

(In the revised Page 10, line 12) Among focal epilepsy, lesional-epilepsy was more common than non-lesional epilepsy (65.1% vs. 34.9%), and temporal lobe epilepsy was more common than frontal lobe epilepsy (40.0% vs. 19.0%).

(In the revised Page 14, line 1) (regardless of lesional or non-lesional epilepsy, temporal lobe epilepsy or frontal lobe epilepsy)

Comment 2:

2) It is not clear the reason for referral for PSG in both subjects with and without epilepsy. It appears that there may be a subject selection bias in both groups that makes it difficult to extrapolate the results. 

Response to comment 2:

Because our study is a retrospective study, there were limitations in accurately identifying the reasons for referral for PSG. The reasons for referral of OE patients were explained in section 4.1, and it was described as a limitation of our study.

(In the revised Page 17, line 10) Reasons for referral included excessive daytime sleepiness, suspected OSA, and characterization of nocturnal spells.

(In the revised Page 19, line 18) The present study had some limitations. First, OE patients were enrolled from an epilepsy clinic; thus, most did not initially appeal to the necessity of a sleep study. This resulted in less severe insomnia, less daytime sleepiness, and less poor sleep quality in OE than in OSA.

Comment 3:

The high percentage of subjects with epilepsy with 2 or more ASMs is surprising, more than expected a priori, and the same for seizure control (subotpimal to say the least). It is also impressive that it does not correctly reflect the population with epilepsy. 

Response to comment 3:

Due to the characteristics of tertiary hospitals, relatively more severe patients may have been included. However, according to a 2007 study based on the Korean National Health Insurance database, the mean number of ASMs prescribed per patient was 1.76 (Reference), which is not significantly different from this study (2.3±1.7). However, acknowledging that it is a single center study as a limitation of this study, I will add the following sentences.

(In the revised Page 19, line 20) Second, since this was a single-center study, the generalizability of our results is limited.

Reference: Lee, Seo-Young, et al. "Prevalence of treated epilepsy in Korea based on national health insurance data." Journal of Korean medical science 27.3 (2012): 285-290.

Comment 4:

It would be interesting NOT only to evaluate according to the number of ASMs but also according to their type (ideally by mechanism of action but at least according to generation). Some of them should have a less neutral impact on the structure of sleep, others could aggravate cardiovascular risk factors which in turn are risk factors for the development of OSA,....

Response to comment 4:

ASMs were classified into four types according to the mechanism of action: voltage-gated ion channels, GABA inhibition, synaptic release machinery and ionotropic glutamate receptors (Reference). In the correlation analysis between this classification and sleep related parameters, ASMs with a mechanism of GABA inhibition showed negative correlation with AHI and AI and positive correlation with ESS, ISI, and BDI. Ionotropic glutamate receptors showed a positive correlation with ESS. However, ASMs acting by modulating voltage-gated ion channels and ASMs acting through interaction with elements of the synaptic release machinery did not show any correlation with sleep-related parameters. These contents were added to the text and tables (2 & 4).

(In the revised Page 5, line 16) The categorization of ASMs according to the mechanism of action referred to previous studies.

(In the revised Page 10, line 16) The most commonly used ASMs are voltage-gated ion channels.

(In the revised Page 14, line 4) Taking ASMs with a mechanism of GABA inhibition showed a negative correlation with AHI (rho = -0.166, p = 0.011) and AI (rho = -0.221, p = 0.001).

(In the revised Page 14, line 9) Taking ASMs having a mechanism of GABA inhibition showed a positive correlation with ESS (rho = 0.149, p = 0.027), ISI (rho = 0.157, p = 0.02), and BDI (rho = 0.219, p = 0.004), and taking ionotropic glutamate receptors showed a positive correlation with ESS (rho = 0.190, p = 0.005).

(In the revised Page 18, line 18) In addition, in our results, taking ASMs with a mechanism of GABA inhibition, such as benzodiazepines, showed a negative correlation with AHI and AI. These drugs are traditionally not recommended in patients with OSA due to the concern of worsening OSA via pharyngeal muscle relaxation, increased apnea duration, and hypoxia [28], but a recent review suggested there was no deleterious effect of the hypnotics on the severity of OSA, as measured by AHI or ODI [29].

Comment 5:

No reference is made to the detection of intercritical epileptiform activity and/or electroclinical seizures in the group with epilepsy. It would be interesting to know whether apneic awakenings are associated with epileptiform abnormalities or not. This would further support the potential use of CPAP as a further added therapy in the course of subjects with epilepsies and OSA.

Response to comment 5:

Among a total of 235 OE patients, 23 patients underwent combined PSG-EEG using 18 channel EEG. Of the 23 patients, 10 had interictal epileptiform activity or electroclinical seizures, and the remaining 13 patients had no abnormalities. Of the 10 patients with epileptiform discharges (EDs), the relationship between EDs and apnea could be analyzed in 7 patients except for 3 with impaired EEG data. However, it was difficult to determine whether there was a relationship between the two factors using this data alone.

Epilepsy type

AHI

No. of EDs

Apnea or hypopnea related EDs

M/57

Focal epilepsy (non-lesional, temporal)

25.9

31

16

M/51

Focal epilepsy (non-lesional, temporal)

7.8

210

1

F/76

Focal epilepsy (non-lesional, temporal)

6.3

876

7

M/44

Focal epilepsy (non-lesional, temporal)

14.6

1145

14

F/37

Focal epilepsy (non-lesional, frontal)

16.2

348

26

M/64

Focal epilepsy (non-lesional, frontal)

45.4

1

1

M/37

Focal epilepsy (non-lesional, temporal)

23.2

2

1

Comment 6:

Ideally, you and myself would have liked to have longitudinal PSG and not just longitudinal sleep questionnaires to assess impact on sleep quality with respect to CPAP use in population with and without epilepsy. In case of technical or economic limitations for its realization, at least I believe that an actigraphy should be postulated. 

Response to comment 6:

We agree with the reviewer's opinion. As the reviewer pointed out, it is ideal to perform longitudinal PSG to analyze the effect of CPAP, but in reality, it is very difficult. As advised by the author, we will actively consider the use of actigraphy in future research.

Comment 7:

The subjects with epilepsy + OSA who complete treatment and follow-up with CPAP are very few and the statement that it reduces the frequency of seizures but do not increase the percentage of seizure-free subjects seems to be related to this sample size and a limited follow-up in time (I think that a one-year analysis could have been of more interest). 

Response to comment 7:

We also acknowledged that the small sample size is a limitation of this study and described it in the limitation section (In the revised Page 19, line 21). In this study, the effects of CPAP use were analyzed on those who used CPAP for at least 6 months, referring to previous papers. Malow et al. (2008) had a treatment phase of 10 weeks. In the study of Vendrame et al. (2011), it was conducted for those who used CPAP for more than 6 months. In the study of Pornsriniyom et al. (2014), the effect of CPAP was analyzed 6 to 12 months after PSG.  

(In the revised Page 19, line 21) In addition, 56 of 134 OE patients prescribed CPAP (41.8%) were seizure-free at baseline, which may have reduced the pool of patients available to assess the effect of CPAP on seizure outcomes. Because ASMs were increased during the CPAP follow-up period in more than one-half of those with OE (45 of 61), the effects of CPAP or ASMs on seizure outcomes were not clearly delineated.

Comment 8:

To evaluate to provide information on intecritical epileptiform activity and the presence of electroclinical seizures and its relationship with AHI in the PSG. 

Response to comment 8:

Among a total of 235 OE patients, 23 patients underwent combined PSG-EEG using 18 channel EEG. Of the 23 patients, 10 had interictal epileptiform activity or electroclinical seizures, and the remaining 13 patients had no abnormalities. However, there was no correlation between interictal epileptiform activity or electroclinical seizures and PSG parameters such as AHI, ODI and AI (see table below). These results cannot be generalized due to the limitation of the sample size, so they are not added to the text. As pointed out by the author, it is necessary to further study this correlation in future studies.

AHI

ODI

AI

ESS

ISI

PSQI

BDI-II

Presence of epileptiform discharges

-0.318

-0.324

-0.202

-0.226

0.187

0.022

0.475

Comment 9:

It would be of interest to evaluate the impact not only on seizure control and self-administered scales of sleep quality and depressive symptoms, but also on other neuropsychiatric symptoms (for example, using the NPI questionnaire) and the impact on global cognitive level (at least with MoCA or MMSE type test) to be performed pre and post-CPAP in both groups.

Response to comment 9:

I agree with the reviewer's opinion. Although this study did not deal with the effect of CPAP on changes in neuropsychiatric symptoms due to the nature of the retrospective study, we agree that neuropsychiatric evaluation is necessary in planning future studies according to the opinion of the reviewer.

Comment 10:

Consider changing the location of Figure 2, which I do not think it would be the most appropriate to place it in the discussion section.

Response to comment 10:

As recommended by the author, the position of Figure 2 was rearranged after Section 3.4 (CPAP adherence and seizure control).

I am aware that this is a retrospective study and that this entails limitations that cannot be rectified, but given the clinical importance of the question posed and the practical applicability of the results of the study, I believe it is essential to demand maximum rigor.

Round 2

Reviewer 2 Report

I am very grateful to the authors for the changes introduced and for the recognition of the limitations of the work, which unfortunately are insoluble. However, in the present form I do consider that it could be of interest for publication.